# ALCC-Glasses: Arriving Light Chroma Controllable Optical See-Through Head-Mounted Display System for Color Vision Deficiency Compensation †

**Ying Tang** [1] , **Zhenyang Zhu** [1] , **Masahiro Toyoura** [2] , **Kentaro Go** [2] , **Kenji Kashiwagi** [3] , **Issei Fujishiro** [4] and **Xiaoyang Mao** [2,5,*]

1    Graduate School of Engineering, University of Yamanashi, Kofu 400-8511, Japan; ty.mail.jp@gmail.com (Y.T.); zhuyamanashi2016@gmail.com (Z.Z.)
2    Department of Computer Science and Engineering, University of Yamanashi, Kofu 400-8511, Japan; mtoyoura@yamanashi.ac.jp (M.T.); go@yamanashi.ac.jp (K.G.)
3    Department of Ophthalmology, University of Yamanashi, Chuo 409-3898, Japan; kenjik@yamanashi.ac.jp
4    Department of Information and Computer Science, Keio University, Yokohama 223-8522, Japan; fuji@ics.keio.ac.jp
5    College of Computer Science, Hangzhou Dianzi University, 1156 2rd street, Hangzhou 310018, China
*    Correspondence: mao@yamanashi.ac.jp
†    This paper is an extended version of our paper published in The 16th ACM SIGGRAPH International Conference on Virtual-Reality Continuum and Its Applications in Industry (VRCAI2018).

**Abstract:** About 250 million people in the world suffer from color vision deficiency (CVD). Contact lenses and glasses with a color filter are available to partially improve the vision of people with CVD. Tinted glasses uniformly change the colors in a user's field of view (FoV), which can improve the contrast of certain colors while making others hard to identify. On the other hand, an optical see-through head-mounted display (OST-HMD) provides a new alternative by applying a controllable overlay to a user's FoV. The method of color calibration for people with CVD, such as the Daltonization process, needs to make the calibrated color darker, which has not yet been featured on recent commercial OST-HMDs. We propose a new approach to realize light subtraction on OST-HMDs using a transmissive LCD panel, a prototype system, named ALCC-glasses, to validate and demonstrate the new arriving light chroma controllable augmented reality technology for CVD compensation.

**Keywords:** color vision deficiency; liquid crystal display; head-mounted display; augmented reality

## 1. Introduction

A retina consists of three types of cone cells, namely L-, M-, and S-cones, which are sensitive to long, medium, and short wavelengths of visible light, respectively. The combination of responses from these three types of cone cells determines the color that is perceived. Genetic disorders can cause one or more cones to be partially or totally nonfunctional, which leads to color vision deficiency (CVD), also known as color blindness [1].

There are 250 million people affected by CVD around the world [2]. Not being able to clearly differentiate certain colors causes those with CVD to face inconveniences, misconceptions, and even dangers in their daily lives. CVD is an inborn deficiency for most of those diagnosed, and there is not yet a cure for the condition; however, contact lenses and glasses with a color filter are commercially available. By applying uniform changes to a user's field of view (FoV), these tools increase the red-green contrast but may filter out other colors. Furthermore, it is reported that wearing tinted

glasses produces an uncomfortable visual experience, and these glasses may potentially impair the user's depth perception [3].

In the field of digital image processing, many methods have been proposed for supporting people with CVD. Some approaches focus on increasing the contrast for pairs of colors by modifying one or both [4–7], while others use additional visual cues, such as patterns, to encode color information [8]. The Daltonization-process-based approaches improve image quality for CVD by increasing or reintroducing lost color contrasts [9–13]. Most of these approaches first use color deficiency simulation methods to compute the decreased or lost information.

Recently, several research groups have proposed to implementing Daltonization using an optical see-through head-mounted display (OST-HMD) [12,13]. OST-HMD provides a computational semitransparent overlay to the environment. By observing the environment through an OST-HMD, where a personal deficiency and environment adapted compensation image are displayed, a CVD user is expected to regain the lost information.

However, color distortion is a problem that needs to be solved to realize the Daltonization process on OST-HMDs. Distortion happens for many reasons, such as the decay of environmental light when passing through an OST-HMD or rendering distortion [14,15]. The most challenging problem, however, is that commercially available OST-HMDs, which are light-additive based, can only emit light that is combined with incoming light from the environment, which makes the perceived image brighter, while the Daltonization process requires some pixel values of the images be decreased.

This paper proposes a novel approach for achieving arriving light control using a liquid crystal display (LCD) for realizing the Daltonization process with an OST-HMD. To validate the feasibility of the idea, a basic prototype consisting of a scene camera, a user-perspective camera, a transmissive LCD panel, and an OST-HMD was presented in [16]. Small lookup tables were used to calibrate the color distortions. The feasibility of the proposed idea was demonstrated with artificially created simple images. This paper proposes ALCC-glasses as an improved prototype system for validating and demonstrating the new arriving light chroma controllable augmented reality technology for CVD compensation. Well designed experiments were conducted to accurately predict the functions required to correct the distortions that occurred during rendering on the OST-HMD and the transmissions through the OST-HMD and the LCD. Algorithms for automatically computing the transparency of the LCD and the images being displayed on an OST-HMD are presented. Various images were used to validate the effectiveness of the proposed approach by comparing the Daltonization process results obtained with and without light subtraction.

Employing an LCD for light reduction is not itself a new idea. Wetzstein et al. [3] developed two prototypes to explore the possibility of using a partially transparent LCD panel for light modulation. One lets users see directly through the LCD, but there may be blur and diffraction artifacts. The other uses extra optical lenses to get the LCD in focus and a beam-splitter set between the LCD and the scene to get an on-axis image so that pixel precision can be ensured, but the structure of the prototype makes miniaturization difficult when placing it in the HMD. Hiroi et al. [17] used a partially transparent LCD panel in their brightness adaptation study. However, their system is for achieving suppression of light in an overexposed area using the LCD and does not allow for individual control of the three color channels.

The contributions of this paper can be summarized as follows:

1. Reproducing color precisely for each pixel is particularly important for Daltonization. The proposed prototype is the first system that can precisely reproduce color by achieving light subtraction with the LCD, while also correcting the color distortion resulting from the transmission of light through an OST-HMD and LCD as well as the rendering on an OST-HMD.

2. A new algorithm is developed for achieving final pixel-wise light chroma control with an OST-HMD while using an LCD to subtract the overall amount of light. In this way, the blur and diffraction artifacts can be alleviated without requiring the LCD to be in focus, which requires incorporating extra devices like optical lenses into the existing system.

3. As a arriving light control technique, the proposed method can be widely used in other augmented reality applications which requiring reduction of individual color channels of the arriving light.

## 2. Related Work

Anagnostopoulos et al. [9] and Badlani et al. [11] put forward the Daltonization process for protanopia (L-cone) and deuteranopia (M-cone) compensation, respectively. As shown in Figure 1, in [9,11], a compensation image, $I_{COM}$, was generated by referring to the original image, $I_{SRC}$, and the CVD simulated image, $I_{SIM}$. The $I_{COM}$ is then superimposed over the $I_{SRC}$ to synthesize the target image, $I_{TAR}$; this image helps individuals with protanopia or deuteranopia identify red and green, which they often struggle to distinguish. In the deuteranopia case [11], $I_{SIM}$ was obtained by removing the information concerning the M-cone from the $I_{SRC}$. The difference between the $I_{SRC}$ and the $I_{SIM}$ was treated as the lost color information, particularly from the G channel, as a result of deuteranopia. Then, the $I_{COM}$ was generated by distributing the lost G channel information across the R and B channels through the following calculations:

$$\begin{bmatrix} R_{COM} \\ G_{COM} \\ B_{COM} \end{bmatrix} = \begin{bmatrix} 1 & 0.7 & 0 \\ 0 & 0 & 0 \\ 0 & 0.7 & 1 \end{bmatrix} \begin{bmatrix} R_{SRC} - R_{SIM} \\ G_{SRC} - G_{SIM} \\ B_{SRC} - B_{SIM} \end{bmatrix}. \tag{1}$$

In this paper, we applied Birch's model [18] to $I_{SRC}$ for the deuteranopia simulation and use the same process as [11] to compute the compensation image. With the appearance of augmented reality (AR) techniques, CVD compensation via wearable devices has become an emerging research topic. Ananto et al. [10] and Tanuwidjaja et al. [13] both proposed CVD compensation solutions with OST-HMDs. In [10,13], the $I_{COM}$ was shown on an OST-HMD superimposed over the realistic scene. However, as already mentioned, the issue of color distortion in OST-HMDs cannot be ignored. In addition, there must be subtraction of arriving light during the Daltonization process, but current OST-HMDs only add light to the background and are not capable of subtracting light.

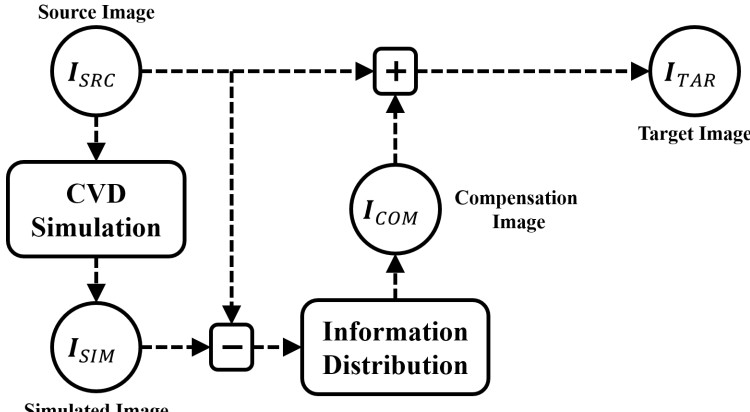

**Figure 1.** Daltonization process.

### 2.1. Color Distortion Correction for the OST-HMD

A lookup table (LUT) that defines the relationship between the input colors and the ones actually observed appears to be one possible way of dealing with the color distortion of an OST-HMD. But, creating a 16-million-color LUT is time-consuming (it may take two to three days), and the OST-HMD would require a considerable amount of memory to store the LUT. To reduce the size, both Sridharan et al. [19] and Hincapié-Ramos et al. [20] utilized a binned-profile method [21] to divide the CIELAB color space into bins with a size of $5 \times 5 \times 5$. To get rid of the LUT, Itoh et al. [14] created a

channel-specific color-correcting model that consists of two portions, a linear transformation matrix and a nonlinear function, according to the physical optics represented in the rendering and perception of an OST-HMD and a user's eyes. Ryu et al. [22] used synthesized images that were captured by a user-perspective camera to estimate real-space colors and performed color correction accordingly. Kim et al. [23] tackled nonlinear color changes through localized regression; they modeled the changes and used hue-constrained gamut mapping for color correction.

## 2.2. Light Modulation for OST-HMDs

In order to make arriving light subtraction possible, Wetzstein et al. [3] developed two prototypes to determine the capability of light modulation using a partially transparent LCD panel. One of the prototypes used an extra optical lenses to get the LCD in focus, and a beam-splitter was set between the LCD and the scene to get an on-axis image so that pixel precision could be ensured. The other prototype let user see directly through the LCD, using an off-axis camera for the scene image. However, the resulting image may suffer from the problems of blur and diffraction.

Hiroi et al. [17] also introduced a partially transparent LCD panel in their brightness adaptation study. In [17], the beam-splitter was set to induce the half-light of a scene to the scene camera for overexposure and underexposure detection. The LCD panel was positioned between the beam-splitter and the OST-HMD to stop a portion of the light from getting through the OST-HMD. They achieved the suppression of light in the overexposed area by utilizing the LCD and the compensation of the underexposed area with the OST-HMD. However, their system processes the three color channels (R,G,B) uniformly and gives no control over the individual color channels.

Itoh et al. [24] introduced an approach for light subtraction based on a spatial light modulator (SLM) with a system that also used an additional optical lens and beam-splitter and provided pixel-level color subtraction. But because their approach also relied on the resolution of the SLM, it also suffered from diffraction because of the pixel grid.

The construction of this paper's proposed prototype is similar to [17]. But, a new algorithm was developed for computing the images to be displayed on an OST-HMD and an LCD to achieve light chroma control, including light subtraction for individual color channels. By achieving final pixel-wise light chroma control with an OST-HMD while using an LCD to subtract an overall amount of light, the blur and diffraction artifacts can be alleviated without requiring getting the LCD in focus, which requires using extra devices, like the optical lenses in [3]. Furthermore, because reproducing color precisely for each pixel is particularly important for Daltonization, the proposed prototype also addresses color correction taking into consideration color distortions resulting from the transmission of light through an OST-HMD and LCD as well as the rendering on an OST-HMD.

## 3. Proposed Method

In this section, we first present the idea of using an OST-HMD and an LCD for realizing the Daltonization process. Next, we introduce the hardware configuration of the ALCC-glasses. Finally, we explain how to control arriving light to achieve the CVD compensation goal.

### 3.1. Realizing the Daltonization Process with an OST-HMD and LCD

As shown in Figure 2, when realizing the Daltonization process with an OST-HMD, the scene image, $I_S$, is the environment a user with CVD is observing, and the target image, $I_{TAR}$, is computed by distributing the lost color information to other color channels as described in Figure 1. This should be equal to the compensated view of the environment observed through the OST-HMD. The displayed compensation image, $\Delta I_T$, is computed based on the difference between $I_S$ and $I_{TAR}$, and because the compensation image may consist of negative pixels, an LCD is used to achieve the subtractive overlay effect.

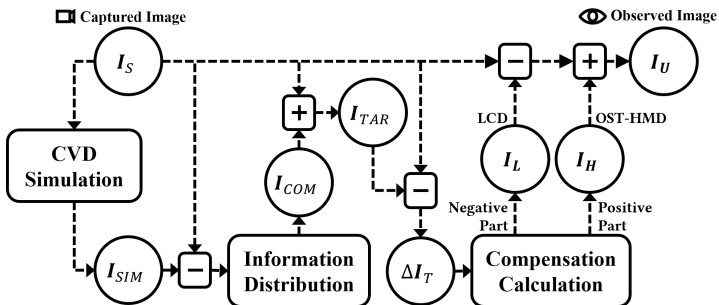

**Figure 2.** Daltonization process with an OST-HMD.

### 3.2. Hardware Configuration

Figure 3 shows the hardware configuration of the ALCC-glasses, which is composed of a scene camera, a beam-splitter, a transmissive LCD panel, an OST-HMD, and a user-perspective camera. The scene camera is used for capturing the scene image, $I_S$, which is required for computing the compensation image, $I_C$. In order to achieve pixel-specific color modification, the image shown on the LCD, the one displayed on the OST-HMD, and the external environment's image must be aligned. We used a 50/50 beam-splitter to separate the environmental light equally into the scene camera and to the LCD and the OST-HMD. The environmental light will decay after passing through the LCD and the OST-HMD. Moreover, nonlinear distortion usually occurs when rendering an image on an OST-HMD. We realized the color calibration by taking into account the decay and distortion in generating the compensation image. For this purpose, a camera was installed at an eye level position in the prototype to capture the user-perspective image, $I_U$; these images were used to collect the data required for the calibration. The user-perspective images were also required for evaluating the effectiveness of the proposed system, which will be introduced in Section 4.

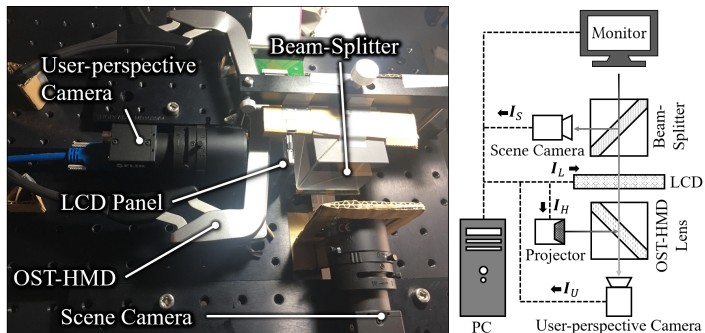

**Figure 3.** Hardware confiuration of the proposed prototype system.

In the current implementation, we used two Blackfly USB color cameras (BFLY-U3-23S6C-C), a Sony LCX017 LCD panel (1024 × 768) with polarizer and controller, an EPSON BT-35E (1280 × 720) OST-HMD, and a 50/50 beam-splitter. We also used an EIZO CS2730 LED monitor to produce environmental light. When the LCD displays a brighter color (e.g., white as $(255, 255, 255)_{RGB}$), it gets more transparent. When it displays a darker color (e.g., black as $(0, 0, 0)_{RGB}$), it gets less transparent and blocks the light. We processed the image taken by the scene camera, and calculated the output for the LCD and OST-HMD on a PC with an Intel Xeon CPU E31245 using a single-threaded naive implementation.

The scene camera and the user-perspective camera both had all automatic adjustments and gamma correction turned off ($\gamma = 1$). For alignment, we adjusted the position of the environmental light (monitor) to a position roughly consistent with, the virtual position of the image rendered on the OST-HMD. Thus, both the environment and the image being shown on the OST-HMD can be seen clearly and pixel-specific color modification becomes possible. To avoid the confusion caused by ambient light, we conducted all experiments in a dark room.

### 3.3. Achieving CVD Compensation with ALCC-Glasses

We ignore the light decrease from the monitor to the scene camera, and assume that the scene camera is observing the scene directly. The environmental light is separated into two rays, but they may not be precisely 50/50. The ray toward the user-perspective camera will then be decreased by the LCD panel and HMD lenses. We use function $f_{decay}$ to represent all of these changes from the scene image, $I_S$, to the user-perspective image, $I_U$ (Figure 4), including the difference of the two rays coming from the beam-splitter. We can control part of the decrease by changing the display image, $I_L$, on the LCD to adjust the transmittance of the LCD. The relationship of $I_L$ and transmittance is represented by $f_{subtraction}$ (Figure 5). Finally, the environmental light will combine with the light emitted by the OST-HMD. When we display an image on the OST-HMD, denoted as $I_H$, the color is distorted for many reasons; this distortion is accounted for by $f_{distortion}$ (Figure 6). With the $I_S$ observed by the scene camera and $I_U$ observed by the user, we have

$$I_U = f_{subtraction}(I_L) \cdot f_{decay}(I_S) + f_{distortion}(I_H). \tag{2}$$

With the three functions in Equation (2) are known, we can compute $I_L$ and $I_H$, which are being displayed by the LCD and the OST-HMD, respectively, using the inverse mapping of the functions. In this subsection, we will first explain how to predict $f_{subtraction}$, $f_{decay}$, and $f_{distortion}$ through experiments, and then explain how to control the overlay enabling CVD users to observe a compensated image through the OST-HMD.

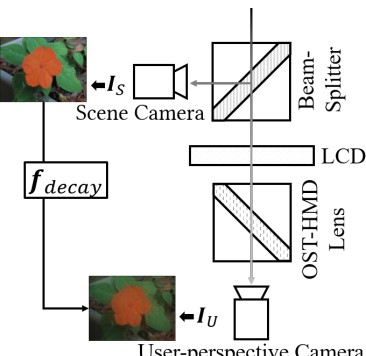

**Figure 4.** $f_{decay}$: The relationship between $I_S$ and $I_U$ when the LCD was set to highest transmittance and the OST-HMD had no output.

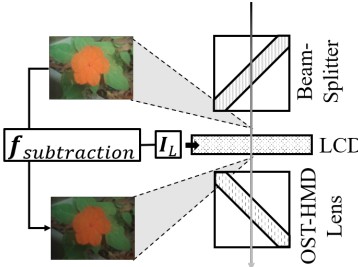

**Figure 5.** $f_{subtraction}$: How much the LCD decreased the incoming light beyond $f_{decay}$.

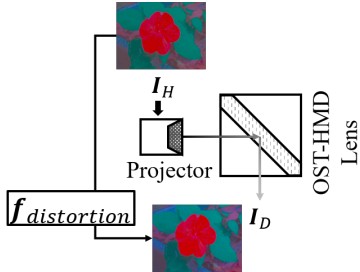

**Figure 6.** $f_{distortion}$: Given an input $I_H$ to the OST-HMD, what appears on the display.

### 3.3.1. Subtraction Function

To predict $f_{subtraction}$, we conducted an experiment by turning off the OST-HMD and displaying an environment color, $C_e$, on the monitor. The image to be sent to the LCD was denoted as $I_{Li}$, a single colored image with color $(i, i, i)_{RGB}$. At this point we found that the transmittance from the LCD was gradient due to a hardware issue. We solved this problem by displaying a linear gradient image instead, which guaranteed that for each $I_{Li}$ ($i \in 0, \ldots, 255$), transmittance from the LCD was even. We changed $i$ from 0 to 255 and recorded the corresponding user-perspective image, $I_{Ui} = (R_{Ui}, G_{Ui}, B_{Ui})_{RGB}$. Figure 7a shows the observed results, with the horizontal axis representing $i$, and the vertical axis standing for the $\dfrac{R_{Ui}}{R_{U255}}$, $\dfrac{G_{Ui}}{G_{U255}}$, $\dfrac{B_{Ui}}{B_{U255}}$ transmittance value as a percentage. Note that the color values may vary by pixels in the user-perspective image, although the environment image is a uniform color image. $R_{Ui}$, $G_{Ui}$, and $B_{Ui}$ are the average value of all pixels in the observed image. As Figure 7a shows, $f_{subtraction}$ is nonlinear. We found the curve could be best fit with a fifth order polynomial. The graphs in Figure 7b are the observed results when the environmental light was set to $(255, 255, 255)_{RGB}$, $(0, 255, 0)_{RGB}$, and $(0, 127, 0)_{RGB}$. All three graphs are very similar; actually, the curves for all of the different $C_e$ values are very similar. We have tested the fitting error of the curves for the different $C_e$ values and found the error increased when the value of any channel in $C_e$ dropped. Therefore, we chose the curve where $C_e = (255, 255, 255)_{RGB}$ for better accuracy in further computing.

Given a target transmittance $t$, we compute the required input, $i$, to LCD by solving the polynomial equation using the Newton–Raphson method. Figure 7a also shows that with the same $t$, there was slight differences in the R, G, and B channels, which we believe was determined by the optical behavior of the LCD's material. Since we cannot change the transmittance of each channel separately, we determined a uniform $i$ from $t$ by solving the fitting equation for the average curve of the three channels using the Newton–Raphson method. Still, the difference of each channel will be considered in further calculations.

Note that in all the experiments in this subsection, with the input as 0, the output still gets a value. This is a result of the camera's light receptor settings. Some types of monitors and OST-HMDs also have the so-called zero-input problem [14]. These constants must be removed when calculating the curve fitting.

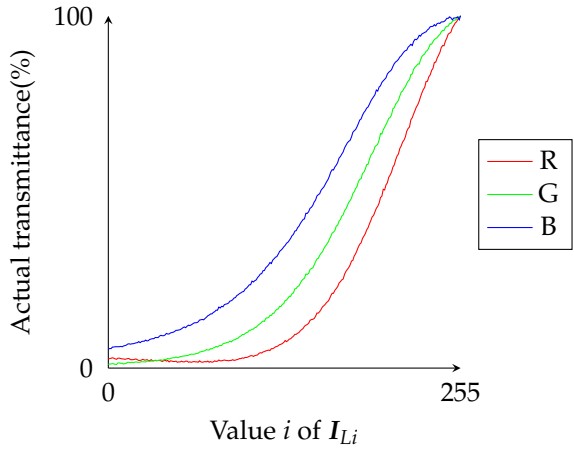

(**a**) Transmittance of LCD when $C_e = (255, 255, 255)_{RGB}$.

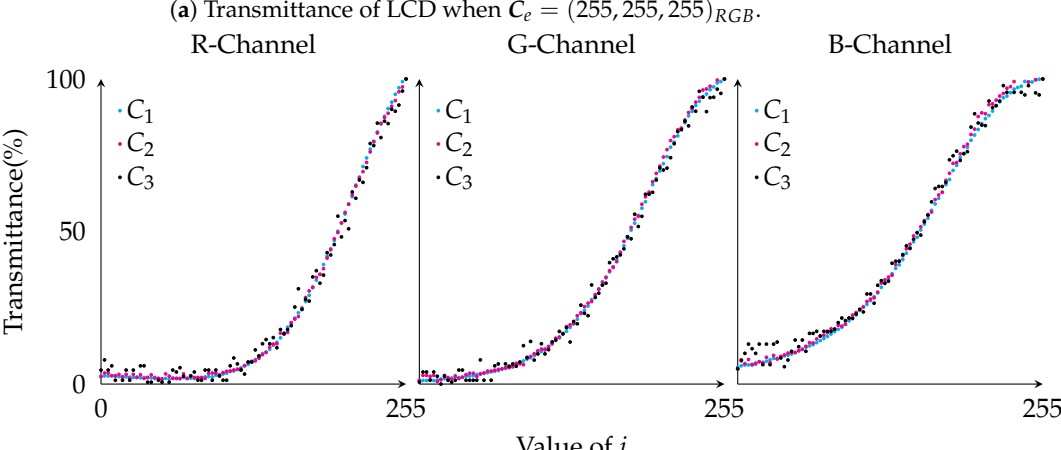

(**b**) Transmittance plotting data of different $C_e$, where $C_1 = (255, 255, 255)_{RGB}$, $C_2 = (0, 255, 0)_{RGB}$, and $C_3 = (0, 127, 0)_{RGB}$.

**Figure 7.** Transmittance curves generated by displaying $I_{Li}$ on the LCD.

### 3.3.2. Decay Function

To estimate $f_{decay}$, we displayed each of the three color channels from 0 to 255 on the monitor with the OST-HMD turned off and set $i$ of $I_{Li}$ to 255 (highest transparency). Since the OST-HMD was off and the LCD was set to its highest transparency, as defined in Equation (2), the $I_S$ and $I_U$ should satisfy $I_U = f_{decay}(I_S)$. In Figure 8, we plot all $(I_S, I_U)$ pairs as points on the 2D space, with $I_S$ along the horizontal axis and the values of $I_U$ in the R, G, and B channels along the vertical axis. The graphs indicate that the relationship between $I_S$ and $I_U$ was linear. As a result of the difference in the color reproduction area of the monitor for each channel, the final observable area of blue light was smaller than red light; hence the line of blue light is shorter than that of red light in Figure 8.

### 3.3.3. Distortion Function

Figure 9a shows the response curves of the user perspective camera when the monitor was turned off and a single-channel color was displayed on the OST-HMD. For the $I_H$, the input image to HMD, we used $I_D$ to represent the distorted image that was observed by the user. As can be seen in Figure 9a, even with single-channel color input, the observation has value in all three channels. This is the side color effect [14]. Figure 9b shows the relationship between side colors and main colors, with the main color value along the horizontal axis and the side color value on the vertical axis. We can observe that

the side colors have a linear relationship to the main colors. Therefore, the main color of each channel is denoted as $R_R$, $G_G$, and $B_B$, and each side color is represented as follows:

$$
\begin{cases}
R_G = a_{R_G} R_R + b_{R_G} \\
R_B = a_{R_B} R_R + b_{R_B} \\
G_R = a_{G_R} G_G + b_{G_R} \\
G_B = a_{G_B} G_G + b_{G_B} \\
B_R = a_{B_R} B_B + b_{B_R} \\
B_G = a_{B_G} B_B + b_{B_G}
\end{cases}.
\tag{3}
$$

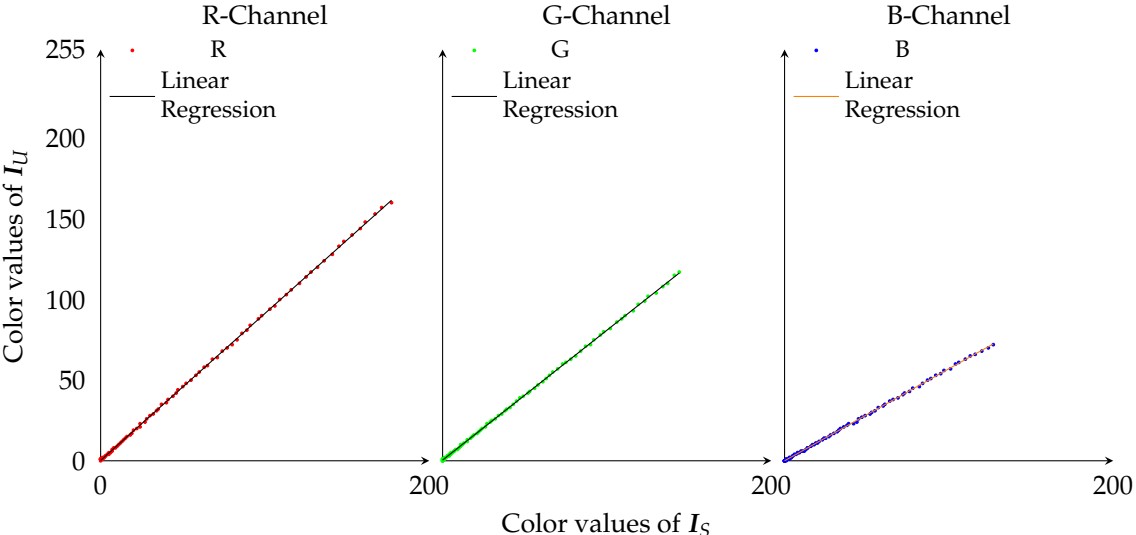

**Figure 8.** Observed color data plot for the decay of light through the system.

Here, $R_G$ and $R_B$ stand for side colors of the R-channel, $G_R$ and $G_B$ for the G-channel, and $B_R$ and $B_G$ for the B-channel. In this way, when $I_H$ is estimated, we only consider the main color components of each channel. Given a distorted image $I_D$, we have

$$
\begin{bmatrix} R_{I_D} \\ G_{I_D} \\ B_{I_D} \end{bmatrix} = \begin{bmatrix} R_R \\ R_G \\ R_B \end{bmatrix} + \begin{bmatrix} G_R \\ G_G \\ G_B \end{bmatrix} + \begin{bmatrix} B_R \\ B_G \\ B_B \end{bmatrix}
$$

with Equation (3), which can be rewritten as

$$
\begin{bmatrix} 1 & a_{G_R} & a_{B_R} \\ a_{R_G} & 1 & a_{B_G} \\ a_{R_B} & a_{G_B} & 1 \end{bmatrix} \begin{bmatrix} R_R \\ G_G \\ B_B \end{bmatrix} = \begin{bmatrix} R_{I_D} \\ G_{I_D} \\ B_{I_D} \end{bmatrix} - \begin{bmatrix} b_{G_R} + b_{B_R} \\ b_{R_G} + b_{B_G} \\ b_{R_B} + b_{G_B} \end{bmatrix}.
\tag{4}
$$

Since $\begin{bmatrix} 1 & a_{R_G} & a_{R_B} \\ a_{G_R} & 1 & a_{G_B} \\ a_{B_R} & a_{B_G} & 1 \end{bmatrix}$ and $\begin{bmatrix} b_{R_G} + b_{R_B} \\ b_{G_R} + b_{G_B} \\ b_{B_R} + b_{B_G} \end{bmatrix}$ are known from the experiment (Figure 9b), we can

compute $\begin{bmatrix} R_R \\ G_G \\ B_B \end{bmatrix}$ by solving Equation (4). With the color $(R_R, G_G, B_B)_{RGB}$, we compute the required

input $I_H$ for the OST-HMD by solving the polynomial equation with the Newton–Raphson method.

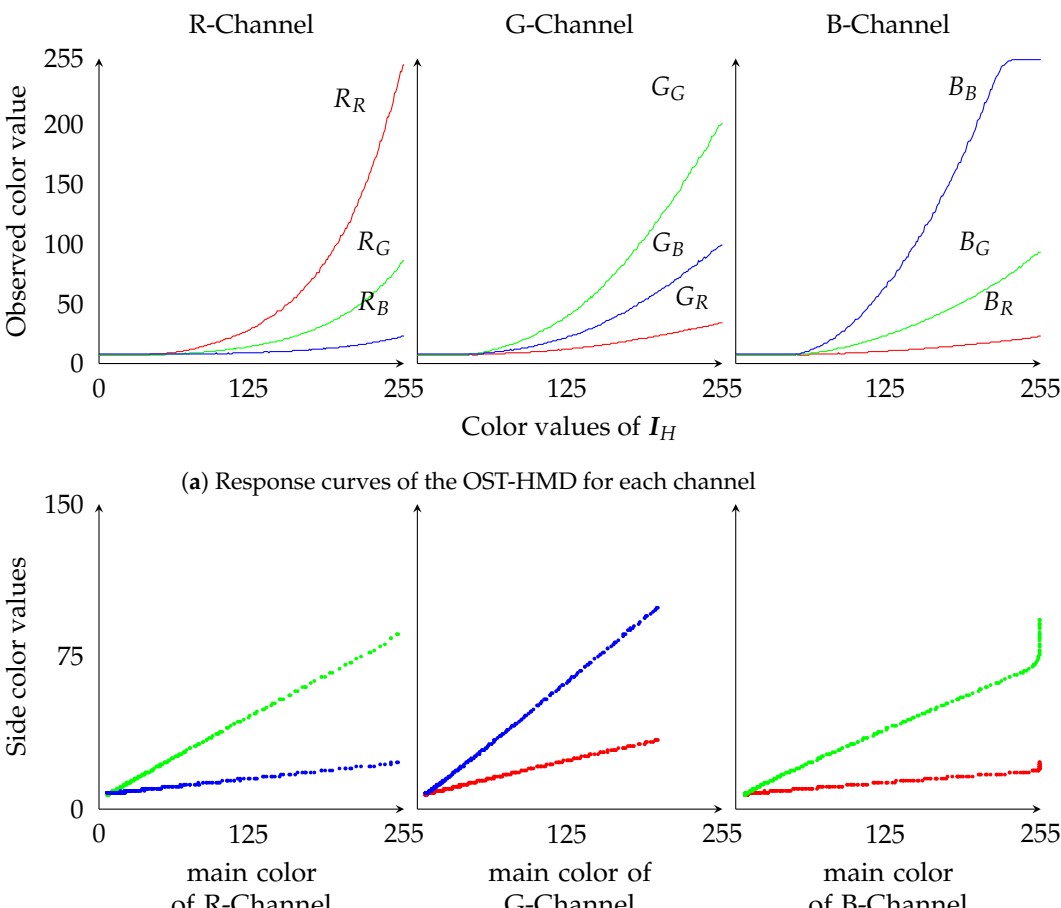

(**a**) Response curves of the OST-HMD for each channel

(**b**) Plotting of side colors to main color for each channel

**Figure 9.** Response curves of the user perspective camera to the OST-HMD rendered image.

### 3.4. Controlling the Overlay to Achieve the Target Image

In this subsection, we explain how to compute $I_L$ and $I_H$ for a given $I_S$ and $I_T$ using Equation (2).

To determine whether light subtraction was necessary and how much light should be subtracted, we calculated the differences between $\Delta I_T$ and $I_T$, the target image computed with the Daltonization process of Equation (1) and $I_U$, the user-perspective image captured with the OST-HMD turned off and the LCD set to $(255, 255, 255)$. If $\Delta I_T = I_T - I_U$ for some pixels is below 0, it means that light subtraction should be performed for those pixels. Because it is difficult to achieve the focus alignment between the LCD, the OST-HMD, and the monitor. we only uniformly controlled the transmittance of the LCD and then achieved the target colors by compensating individual color channels for each pixel using the OST-HMD. For the computation strategy of the transmittance, we propose two principles, the lowest transmittance principle (Principle I), and the histogram peak principle (Principle II). For both Principles I and II, each channel of all the pixels will be judged regarding the need for light subtraction. Consider the set of pixels $P_{negtive}$; arbitrary pixel $p_i$ has at least one negative channel value in $\Delta I_{Ti}^2$, so it requires light subtraction. Then, the light subtraction rate (LSR) and the corresponding transmittance $t$ of the channel are calculated as

$$LSR(C, p_i) = \frac{\Delta C_i}{C_{Ui}}$$

$$t(C, p_i) = 1 - LSR(C, p_i).$$

where $C \in \{R, G, B\}$ and $\Delta C_i < 0$. In Principle I, the lowest transmittance of all pixels is set as the transmittance of the LCD panel; channels of pixels that are over subtracted by the transmittance $t$ are supposed to be compensated for by the OST-HMD. In Principle II, the pixel number of each channel

and transmittance is counted as the height of the bin in the two-dimensional histogram $H$. Each bin of the histogram, $H(C, k)$, is combined by a chromaticity channel $c$ with $(R, G, B)$ and $LSR(k)$, which is in hundredths.

$$H(C, k) = \sum_{p_i \in \{R,G,B\}} \begin{cases} 1, LSR(C, p_i) = k \\ 0, otherwise \end{cases}.$$

Then, we sum up the histogram $H$ by chromaticity channel and get a one-dimensional histogram $\hat{H}$,

$$\hat{H}(k) = \sum_{C \in \{R,G,B\}} H(C, k).$$

Finally, the $k$ corresponding to the maximal number of pixels of $\hat{H}$ is set as the target LSR, and $f_{subtraction}$ is set as

$$f_{subtraction}(I_L) = 1 - \underset{k}{argmax}\,\hat{H}(k).$$

In theory, Principle I should be more correct. However, we adopted Principle II because of the limitations of the OST-HMD, which will be elaborated on in the discussion section. $f_{subtraction}(I_L)$ is computed as

$$f_{subtraction}(I_L) = 1 - \underset{c,i}{max}(\frac{\left|\Delta I_{Ti}^c\right|}{I_{U_i}^c}), \Delta I_{Ti}^c < 0,$$

where $t = \frac{\left|\Delta I_{Ti}^c\right|}{I_{U_i}^c}$ stands for the subtraction required by the $c$ channel ($c = R, G, B$) of pixel $p_i$. Then, the controlling image $I_L$ of the LCD is computed by the method described in Section 3.3.1, and a new user-perspective image $I_U'$ is estimated as

$$I_U' = f_{subtraction}(I_L) \cdot I_U.$$

If $\Delta I_T$ for all pixels is greater than or equal to 0, $I_L$ is set to $(255, 255, 255)$, no subtraction is required, and $I_H$ can be calculated directly following the difference calculation.

For computing $I_H$, the difference between $I_T$ and the estimated user-perspective image $I_U'$ denoted as $\Delta I_T'$, is used to compute how much each channel of each pixel should be compensated for by the OST-HMD. This means the distorted image $I_D$ emitted by the OST-HMD should equal $\Delta I_T'$; therefore, we can use Equation (4) to get $(R_R, G_G, B_B)_{RGB}$ and then compute the required input, $I_H$, for the OST-HMD as described in Section 3.3.3.

## 4. Result and Evaluation

In order to verify the effectiveness of the proposed method, we conducted an experiment to compare our results with the existing method. Both [12,13] have the same purpose as ours, that is, achieving color compensation by displaying overlay image on OST-HMD. Both methods do not support the light subtraction. We choose to compare with [13] as it also uses daltonization so that we can confirm the effect of the light subtraction while eliminating other factors caused by the difference of compensation methods.

Given the $I_S$ captured by the scene camera from the monitor, the $I_T$ is equal to the target image for deuteranopia, which is obtained by adding a compensation image computed with Equation (1) to the $I_S$. Denoting the image provided to the user with our prototype system as $I_U^{our}$ and with the existing method [13] as $I_U^{ex}$, we compared the similarities of $I_U^{our}$ and $I_U^{ex}$ to the computed target image $I_T$. $I_U^{our}$, which is captured with the user-perspective camera, is generated by first computing the compensation image with Equation (1) using $I_S$ and $I_T$, and then by computing the LCD panel controlling image $I_L$ and the OST-HMD overlay image $I_H$ with the estimated $f_{subtraction}$, $f_{decay}$, and $f_{distortion}$ using Equation (2). $I_U^{ex}$, which is also captured by a user perspective camera, is generated by displaying the compensation image generated with Equation (1) directly on the OST-HMD with the LCD removed

from the system. For those pixels with negative values in the compensation image, we changed the negative value to 0.

With the LCD installed, the color area that can be observed by the user is actually smaller than the area without the LCD because of the decay and the material color of the LCD. As shown in Figure 10, with the compensation for both the OST-HMD and the LCD, our method (Figure 10, 3rd column) appeared to be more similar to the target images (Figure 10, 2nd column) and showed enhanced contrast as compared to the existing method (Figure 10, 4th column) for the areas where contrast loss tends to occur with deuteranopia. The resolution of all images is $400 \times 300$, and the average processing time was 30.91 milliseconds.

Furthermore, as a quantitative evaluation, we used the root mean square (RMS) error metric proposed in [24] to compare our method with the existing method. The RMS evaluates the consistency of the local contrast between a pixel $q_i$ and its randomly picked neighboring pixels ($k$ in total) $S_{qi} = \{q_i, q_{i+1}, ..., q_{i+k-1}\}$ in the test image with that of the corresponding pixel $p_i$ and the neighboring pixels $S_{pi} = \{p_i, p_{i+1}, ..., p_{i+k-1}\}$ in the reference image,

$$RMS(q_i) = \sqrt{\frac{1}{k} \sum_{s=j}^{j+k-1} (\frac{(p_i - p_s) - (q_i - q_s)}{160})^2},$$

where the constant 160 was introduced to normalize the index value to the range of [0,1]. We applied the RMS to the CVD simulation image of the target image, the simulation result using the proposed method, and the simulation result using the existing method. The simulation result of the original image was used as a reference image.

By contrasting the results of the proposed method to the results of the existing method (shown in Table 1), it is clear that the proposed method's results are superior other than for group C. Since the image of group C consists of more high frequency on the background, the contrast loss caused by such blurring artifacts, which will be discussed in the next section, becomes more dominant even though the foreground object that is supposed to be barely visible with deuteranopia becomes much clearer than in the existing method.

**Table 1.** RMS results for each image; a smaller RMS represents a bigger contrast.

| RMS　　　Method<br>Group | Ideal Daltonization | Proposed Method | Existing Method |
|---|---|---|---|
| Group A | 95.65 | 109.59 | 138.46 |
| Group B | 118.77 | 120.48 | 154.36 |
| Group C | 58.29 | 128.13 | 103.07 |
| Group D | 71.22 | 94.78 | 105.88 |
| Group E | 144.20 | 172.05 | 175.25 |

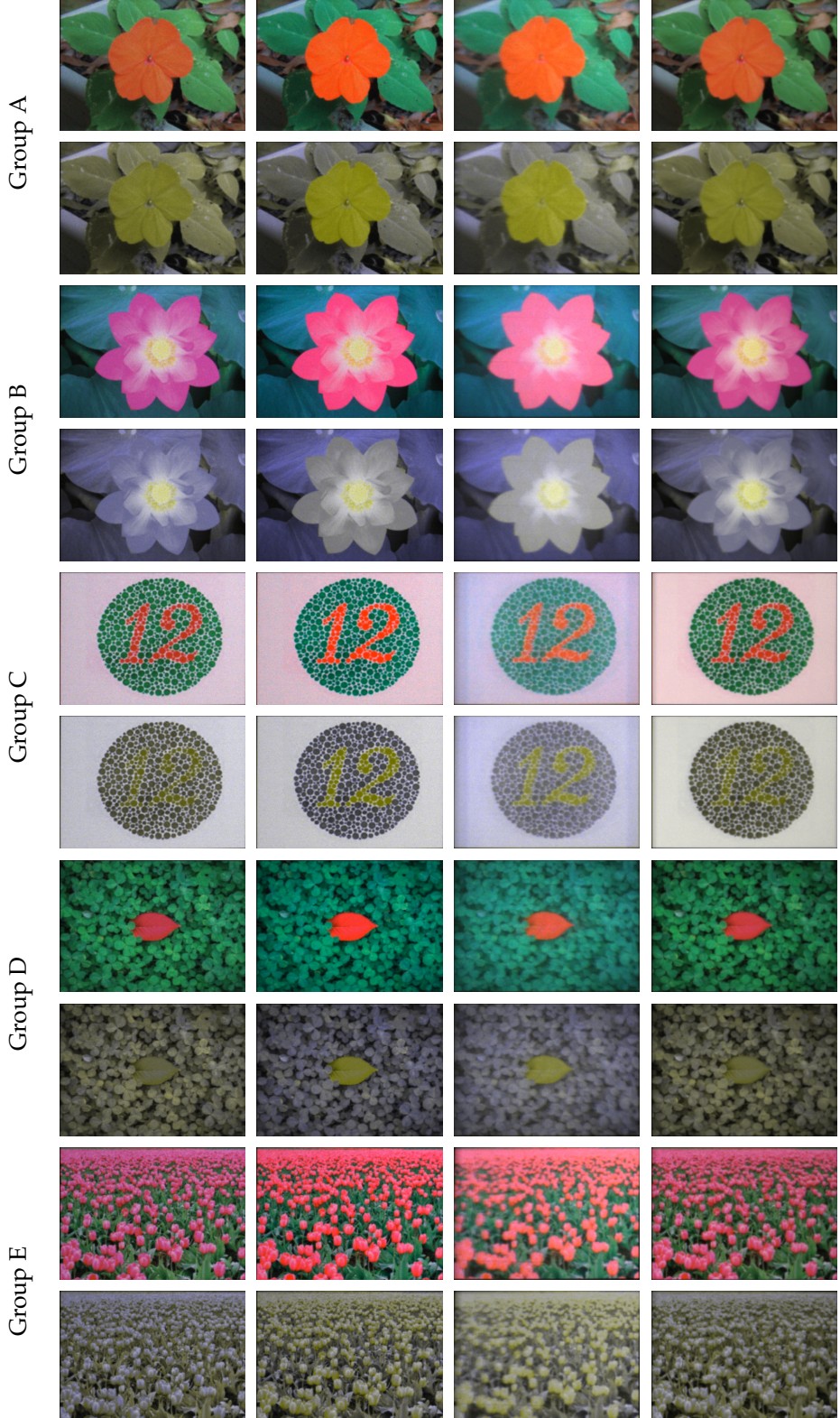

**Figure 10.** Pictures in odd rows from left to right are: images captured by the scene camera, $I_S$; the ideal compensation results, $I_T$, by applying the Daltonization process to $I_S$; the result using our proposed method; and the result using the existing method [13]; pictures in even rows are from the deuteranopia simulation following the same order of methods as given above.

## 5. Discussion

By introducing a beam-splitter and an LCD panel into the system, complexity is increased. The compensation stability is greater or lesser, depending on the specification of each optical unit. The light is decreased about 50% by the beam-splitter and will be further decreased by the LCD and the OST-HMD. The light throughput of the system will be less than 10%. However, although we achieved light compensation using an HMD, the extra structure makes miniaturization harder. In further studies, we are considering using a stereo scene camera and simulating the on-axis image from that input instead of using a beam-splitter.

In this study, we ignored the focus problem and forced the user perspective camera to focus on the monitor and the OST-HMD. This is one of the major reasons for the blurring of the artifact observed in Figure 10, although the artifact has been reduced from that of the existing methods [13] by using the OST-HMD to compensate colors for individual pixels. This problem needs to be better addressed to use our system in a real-life application. When seen through the LCD panel, there are some diffraction effects from the pixel grid, which is inherently caused by the property of the LCD panel used (bbs bild- und lichtsysteme GmbH. http://www.bbs-bildsysteme.com/) and might be improved by decreasing the resolution of the LCD since that does not affect the results of our method.

Due to the focus problem, we can only subtract light uniformly for an entire image because pixel-wise alignment cannot be made between the LCD and the other devices. The uniform amount of light to be subtracted is decided by computing the histogram of needs percentage for the three colors of all pixels and selecting the needs percentage of the highest bin in the histogram. Although we compensated the over- or under-subtracted areas with the OST-HMD, errors may remain. Figure 11a visualizes the need for subtraction with intensity. Figure 11b,c visualize the errors that are computed as the difference between the resulting image and the target image. When $t = 33$, which is the highest needs percentage corresponding to the bright area in Figure 11a, we can get a contrast enhanced image as shown in Figure 11e, but there are errors in the green channel, on the flower area. This is likely because these pixels require a very small value for the green channel and the zero-input effect mentioned in Section 3.3.3 prevented that channel from reaching zero. When $t = 100$, with the positive compensation by the OST-HMD and color calibration, the result still had a fair contrast but lost some of the details compared to the result obtained with $t = 33$ (the yellowish part on the leaves). This indicates that a good transmittance selection policy may reduce the number of error pixels.

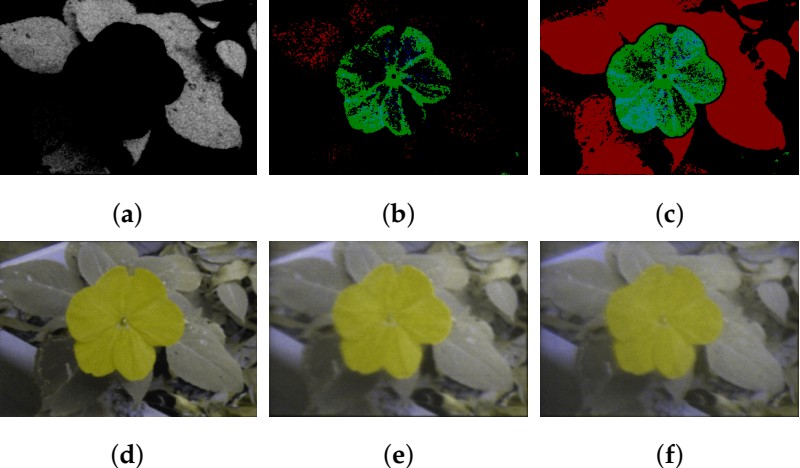

|          |          |          |
| :------: | :------: | :------: |
| (**a**)  | (**b**)  | (**c**)  |
| (**d**)  | (**e**)  | (**f**)  |

**Figure 11.** (**a**) Bright pixels are ones that need subtraction; (**b**) error pixels when $t = 33$, color stands for the corresponding color channel; (**c**) error pixels when $t = 100$; (**d**) simulation of $I_T$; (**e**) Simulation of $I_U$ when $t = 33$; and (**f**) Simulation of $I_U$ when $t = 100$.

By making it possible to calibrate and compensate for incoming light and making the image in the user's FoV reach a target image, the proposed technology can be combined with other compensation

methods for CVD or even other augmented reality applications using OST-HMD where arriving light control is necessary.

**Author Contributions:** Conceptualization, X.M.; methodology, Y.T., Z.Z., M.T., K.G., K.K., I.F. and X.M.; software, Y.T.; validation, K.K. and X.M.; formal analysis, Y.T.; investigation, Y.T. and K.K.; resources, X.M.; data curation, Y.T.; writing—original draft preparation, Y.T. and Z.Z.; writing—review and editing, X.M.; visualization, I.F.; supervision, M.T. and X.M.; project administration, X.M.; funding acquisition, X.M. All authors have read and agreed to the published version of the manuscript.

**Funding:** This study was funded by JSPS Grants-in-Aid for Scientific Research Grant No. 17H00738.

**Conflicts of Interest:** The authors declare no conflict of interest.

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
