# Peer review of "ALCC-Glasses: Arriving Light Chroma Controllable Optical See-Through Head-Mounted Display System for Color Vision Deficiency Compensation†"

_applsci, doi:10.3390/app10072381_

Round 1
Reviewer 1 Report
This paper described a head-mounted display system for the compensation of color vision deficiency. The proposed daltonization process and hardware configuration add good amount of advantage on top of the conventional daltonization process. The reported results look promising in terms of RMS values. It will be more convincing if more existing methods can be added for comparative study because the current one is a >5 years work.
Author Response
Thank you for your valuable comments. As mentioned in the paper, the recent researches that have similar device mechanism or purpose to our study are [12] and [24]. While [12] uses almost the same approach to our comparisons (instead of the daltonization process, they use three other algorithm), and [24] does not use OST-HMD. To the best of our knowledge, there is no other research has same purpose of ours.
Thus, as a supplement, we changed the first paragraph of section 4 to:
In order to verify the effectiveness of the proposed method, we conducted an experiment to compare our results with the existing method. Both [12] and [13] have the same purpose as ours, that is, achieving color compensation by displaying overlay image on OST-HMD. Both methods do not support the light subtraction. We choose to compare with [12] as it also uses daltonization so that we can confirm the effect of the light subtraction while eliminating other factors caused by the difference of compensation methods.
[12] Langlotz, T.; Sutton, J.; Zollmann, S.; Itoh, Y.; Regenbrecht, H. ChromaGlasses: Computational Glasses for Compensating Colour Blindness. Proceedings of the 2018 CHI Conference on Human Factors in Computing Systems. ACM, 2018, p. 390.
[13] Tanuwidjaja, E.; Huynh, D.; Koa, K.; Nguyen, C.; Shao, C.; Torbett, P.; Emmenegger, C.;Weibel, N. Chroma: a wearable augmented-reality solution for color blindness. Proceedings of the 2014 ACM International Joint Conference on Pervasive and Ubiquitous Computing. ACM, 2014, pp. 799–810.
[24] Itoh, Y.; Langlotz, T.; Iwai, D.; Kiyokawa, K.; Amano, T. Light Attenuation Display: Subtractive See-Through Near-Eye Display via Spatial Color Filtering. IEEE Transactions on Visualization and Computer Graphics 2019, 25, 1951–1960.
Reviewer 2 Report
Dear Authors,
your work is very interesting and well done I have only few suggestions:
Line 62 – there is a typo in ‘er al.’
Line 156 – in the word ‘naïve’, ‘i’ has two dots
Line 263 – there is ‘beam-slitter’ and should be beam-splitter’
Figure 10 is described after figure 11, maybe it would be better to consider the change in figures or description order.
Best wishes
Author Response
Thank you for your valuable comments. We have fixed the mistakes mentioned in the comments:
Point 1: Line 62 – there is a typo in ‘er al.’
Response 1: Changed 'Hiroi er al.' to 'Hiroi et al.' in line 62.
Point 2: Line 156 – in the word ‘naïve’, ‘i’ has two dots
Response 2: Changed 'naïve' to 'naive' in line 156.
Point 3: Line 263 – there is ‘beam-slitter’ and should be beam-splitter’
Response 3: Changed 'beam-slitter' to 'beam-splitter' in line 263.
Point 4: Figure 10 is described after figure 11, maybe it would be better to consider the change in figures or description order.
Response 4: We have changed the order of two figures.